# Optimizing NMF Clustering Loss Functions Using Relational Table Flow of Information for Hidden Knowledge Discovery

## Abstract

In the era of artificial intelligence, there is an acceleration of high-quality inference from the fusion of data. We propose a novel solution to the linking challenge associated with higher-order features. We have fundamentally linked together and factored database tables using custom NMF loss functions and expect this paradigm and those related to it to produce many new insights. This algorithm extends multi-view clustering, star based schema and heterogeneous information networks by adding complementary and consensus information across linked views of each data point. While MultiNMF, multi-view clustering and star based schema incorporates the first link and heterogeneous information networks focus on the graph there exist troves of data that still are not incorporated. Basically we extend heterogeneous information network analysis to consider not just the edges, nodes and the embeddings of these nodes and edges, but we use the familiar keyed relationships of relational databases in their tabular format. This optimizes for the structure of databases more specifically than previous works by recreating a relational database from all of its tabular factors. To our knowledge, a general artificial intelligence method to factor all tables in a chain has not been done before. We call this chained NMF or chained view clustering and give the algorithms to perform multiplicative updates and the general solution that can be solved using automatic differentiation such as JAX. We demonstrate how the equations can be interpreted on synthetic data and how information flows through the links. As a proof of concept on real data, we incorporate word vectors using the method on an authorship clustering dataset.

## 1 Introduction

Until recently, clustering algorithms typically use either a table or at most two relational tables to analyze data. However there is a demand to fuse larger quantities of data for greater algorithmic success, we will show one reasonable method to do so. We adopt a simple notation of row $\times$ column to represent a relational table where the rows are items and the columns are features. Now we will begin with an example, community detection based topic modeling using document clustering on social media derived data with NMF, our table is a term $\times$ document table. In this example word vectors can be incorporated by adding a word vector table using MultiNMF via a word $\times$ word vector table. However this leaves large amounts of available data out of the factoring. We propose chaining several tables together for instance: hashtag $\times$ word, word $\times$ location, location $\times$ author, author $\times$ topics, as seen in Figure 1. From this information, we may identify communities that are more broadly informed than simply the words present in the social media posts. At the heart of artificial intelligence is extracting previously unknown and potentially useful information from all available data. However, the problem is that this extent of data fusion using higher-order features has not been done for NMF in prior works.

Although we focus on NMF using multiplicative update rules, MUR, Seung & Lee (2001) it is entirely possible to define a joint loss function, J, as seen in the three subsections below using automatic differentiation such as JAX, Torch, or TensorFlow. The reason to use MU over convex optimizers, alternating nonnegative least squares, and automatic differentiation is the reduced computational load and higher empirically observed quality of the results. However, we have implemented a MUR

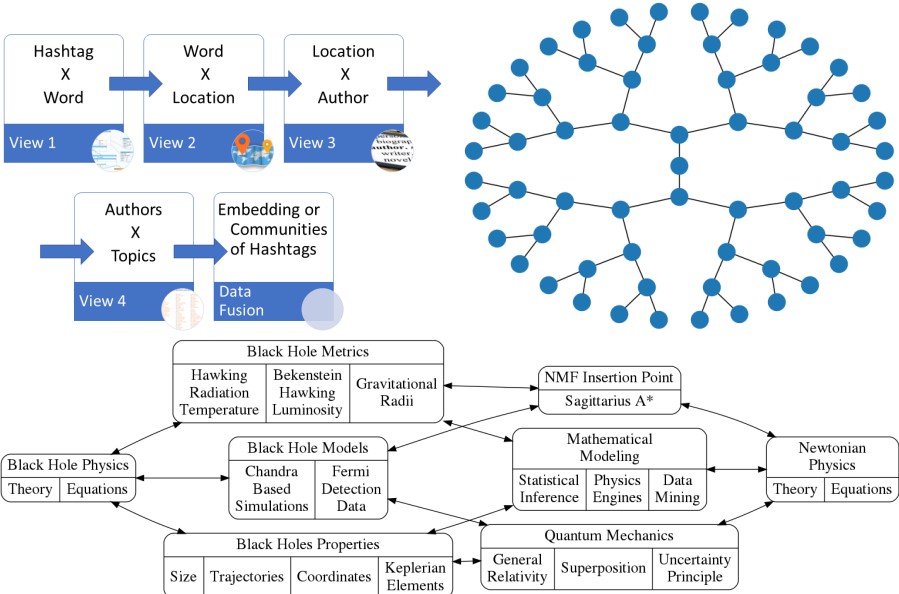

Figure 1: Chaining together views to gain a whole view of multiple interrelated datasets with the only requirement that features are keyed by other features. While a linear chain is shown in the left of the figure, it can actually be a radial branching chain as seen in the right of the figure. It is shown symmetric, but there is no requirement for symmetry. This method differs from graph embeddings because entire relational tables factors can reconstruct the nodes in the graph. (Bottom) Depicted is a fictitious relational dataset model relating to black holes where arrows represent primary keys that are not provided. We propose performing analysis at the NMF insertion point Sagittarius A* analyzing at the datapoint level, this novel approach is different than analyzing the graph structure with graph embeddings or only the immediate neighboring tables. Clusters, factors and other NMF outputs are not shown, but only the input structure in all three of the subfigures.

based algorithm that uses automatic differentiation that is faster and provides comparable results to MURs. To appreciate the challenges that arise in chained view NMF, we must consider that for multi-view NMF, we have a much easier to solve MUR and can easily incorporate more significant numbers of views than we could incorporate many linked relations. MultiNMF is based upon the precept that the primary key has features. The difference between multi-view clustering and traditional table based clustering is that the first treat features as sets of similar types, and the latter treats all features equally. However, linking relations cannot be captured in either form of clustering directly nor graph based embeddings such as heterogeneous information networks, this difference is this work's main contribution.

## 2 TYPES OF NMF

Here we present the NMF objective functions and its corresponding MUR. Next we show the Multiview NMF that factors multiple matrices related by the primary datatype. In this type we give the MUR rule and its generalization to any number of primary datatype interrelated matrices. Finally, we illustrate chained view NMF that allows linking of matrices in the same way as a relational database, i.e. feature matrices of features. In linked view we explored the analytic solution but due to computational hurdles we additionally defined an automatic differentiation to ascertain the positive and negative terms of the joint loss function thus allowing us to perform the MUR.

### 2.1 VANILLA NMF

All division is elementwise and $\odot$ is elementwise multiplication, finally $\times$ represents a relational dataset composed of rows whose type is before the $\times$ and the features whose type is after the $\times$. An

example of Vanilla NMF is a matrix of hashtags $\times$ tf-idf of words and yielding two matrices, W and H. W is the hashtag communities or embeddings and H is the community word usage.

$$Loss = \|\mathbf{V} - \mathbf{WH}\|_F^2 \tag{1}$$

$$\mathbf{W} \leftarrow \mathbf{W} \odot \frac{\mathbf{VH}^T}{\mathbf{WHH}^T} \tag{2}$$

$$\mathbf{H} \leftarrow \mathbf{H} \odot \frac{\mathbf{W}^T\mathbf{V}}{\mathbf{W}^T\mathbf{WH}} \tag{3}$$

and for KL Divergence from Burred (2014):

$$Generalized\ D_{KL}(P\|Q) = \sum_i \left[ P(i) \log \frac{P(i)}{Q(i)} - P(i) + Q(i) \right] \tag{4}$$

$$Loss = D_{KL}(\boldsymbol{V}\|\boldsymbol{WH}) = \sum_{i,j} \left[ V_{i,j} \log \frac{V_{i,j}}{WH|_{i,j}} - V_{i,j} + WH|_{i,j} \right] \tag{5}$$

$$\mathbf{W} \leftarrow \mathbf{W} \odot \frac{\frac{\mathbf{V}}{\mathbf{WH}}\mathbf{H}^T}{\sum_i \boldsymbol{H}_{i,:}} \tag{6}$$

$$\mathbf{H} \leftarrow \mathbf{H} \odot \frac{\boldsymbol{W}^T \frac{\boldsymbol{V}}{\boldsymbol{WH}}}{\sum_j \boldsymbol{W}_{:,j}} \tag{7}$$

## 2.2 MultiNMF

For the case of 2 matrices such as hashtag $X$ word and hashtag $X$ location representing $V_A$ and $V_B$:

$$Loss = \|\mathbf{V_A} - \mathbf{WH_A}\|_F^2 + \|\mathbf{V_B} - \mathbf{WH_B}\|_F^2 \tag{8}$$

$$\mathbf{W} \leftarrow \mathbf{W} \odot \frac{\mathbf{V}_A\mathbf{H}_A^T + \mathbf{V}_B\mathbf{H}_B^T}{\mathbf{WH}_A\mathbf{H}_A^T + \mathbf{WH}_B\mathbf{H}_B^T} \tag{9}$$

$$\mathbf{H}_A \leftarrow \mathbf{H}_A \odot \frac{\mathbf{W}^T\mathbf{V}_A}{\mathbf{W}^T\mathbf{WH}_A} \tag{10}$$

$$\mathbf{H}_B \leftarrow \mathbf{H}_B \odot \frac{\mathbf{W}^T\mathbf{V}_B}{\mathbf{W}^T\mathbf{WH}_B} \tag{11}$$

for the general case we have the following equation:

$$Loss = \|\mathbf{V_A} - \mathbf{WH_A}\|_F^2 + \|\mathbf{V_B} - \mathbf{WH_B}\|_F^2 + ... + \|\mathbf{V_Z} - \mathbf{WH_Z}\|_F^2 \tag{12}$$

$$\mathbf{W} \leftarrow \mathbf{W} \odot \frac{\mathbf{V}_A\mathbf{H}_A^T + \mathbf{V}_B\mathbf{H}_B^T + ... + \mathbf{V}_Z\mathbf{H}_Z^T}{\mathbf{WH}_A\mathbf{H}_A^T + \mathbf{WH}_B\mathbf{H}_B^T + .... + \mathbf{WH}_Z\mathbf{H}_Z^T} \tag{13}$$

$$\mathbf{H}_A \leftarrow \mathbf{H}_A \odot \frac{\mathbf{W}^T\mathbf{V}_A}{\mathbf{W}^T\mathbf{WH}_A} \tag{14}$$

$$\mathbf{H}_B \leftarrow \mathbf{H}_B \odot \frac{\mathbf{W}^T\mathbf{V}_B}{\mathbf{W}^T\mathbf{WH}_B} \tag{15}$$

$$... \tag{16}$$

$$\mathbf{H}_Z \leftarrow \mathbf{H}_Z \odot \frac{\mathbf{W}^T\mathbf{V}_Z}{\mathbf{W}^T\mathbf{WH}_Z} \tag{17}$$

## 2.3 OUR PROPOSED WORK - CHAINED VIEW NMF

What if the relational table that you would like to incorporate does not have the same type of rows as $\mathbf{V}$. In star based schema and MultiNMF if $\mathbf{V}$ is hashtag rows and word frequency columns then you must use a hashtag row table, such as hashtag rows and location column. In chained view the requirement is relaxed such that any table can be used that has the same column type as $\mathbf{V}$ for instance $\mathbf{V}_B$ has word rows and word vector columns this is possible. What becomes apparent is that by relaxing this requirement then any number of possible tables can be joined together for instance: hashtag rows to TF-IDF columns table can join a word row by location column and then join a location row to a author column table and finally a author row by topics column table. Because this follows the principle of foreign keys in a relational database, it is possible to directly use this algorithm to take a whole database and gives clusters based on every datapoint in it, this goes beyond a graph based topological view or embedded view of a table such as heterogeneous information networks. This is in effect a feature of features solution using higher order features. Using the example from before in the text we formulate the problem as such:

$$
\begin{aligned}
Loss = &\|\mathbf{V}_A - \mathbf{W}\mathbf{H}_A\|_F^2 + \\
&\|\mathbf{V}_B - \mathbf{V}_A^T\mathbf{W}\mathbf{H}_B\|_F^2 + \\
&\|\mathbf{V}_C - \mathbf{V}_B^T\mathbf{V}_A^T\mathbf{W}\mathbf{H}_C\|_F^2 + \\
&\|\mathbf{V}_D - \mathbf{V}_C^T\mathbf{V}_B^T\mathbf{V}_A^T\mathbf{W}\mathbf{H}_D\|_F^2 + \\
&... \\
&\|\mathbf{V}_Z - \mathbf{V}_Y^T\mathbf{V}_X^T...\mathbf{V}_B^T\mathbf{V}_A^T\mathbf{W}\mathbf{H}_Z\|_F^2
\end{aligned}
\tag{18}
$$

Again using the notation of a table as row$\times$column rather than crossproducts, we have W is the hashtag communities of the database, $\mathbf{V_A}$ through$\mathbf{V_D}$ are hashtag $\times$ tf-idf, word $\times$ location, location $\times$ author and finally author $\times$ topics respectively. Interestingly this also produces an embedding of the database for hashtags. It may be important to decrease the loss assigned to increasingly complex chains so they do not contribute to W the same as the most proximal tables. Alternatively one may solve for the last term in the loss function and by doing so incorporate all of the data matrices. Solving for the last term does have one drawback, it disproportionately weighs all data matrices equal, this may not be the intent. It may be necessary to experiment with different weightings of the terms by scaling the matrices proportional to their weighting, for instance taking the elementwise product of a number between 0 and 1 with the data matrix to be weighted.

We will begin with the MUR rule using auto differentiation. We must take the loss function and convert it into individual terms that can then be entered into an automatic gradient solver. For instance Frobenius norms produce: $Tr((X - WH)^T(X - WH))$, performing FOIL and using matrix properties you then have terms that can be solved by the autodifferentiation. The reason Frobenius norms are not used directly by the gradient solver is that they do not create the positive and negative terms individually that are needed for the MUR algorithm.

---

**Algorithm 1** Function: Multiplicative Update Auto Differentiation

---

**Input:** Joint Loss Function
**Output:** Matrix to Multiply by W, H, H2, H3, ..., Hn during update
    *Initialisation* : Define Positive and Negative Terms of the Joint Loss Function
    *grad* : Providing the Gradients through Auto Differentiation i.e. JAX for Specified Variable
 1: Gradients = Absolute Value of All Terms after Application of *grad*
 2: Numerator = Sum of All Negative Terms' Gradients
 3: Denominator = Sum of All Positive Term's Gradients
 4: Denominator += Epsilon of Machine
 5: **return** Numerator / Denominator

---

$$
Loss = \|\mathbf{V}_A - \mathbf{W}\mathbf{H}_A\|_F^2 + \|\mathbf{V}_B - \mathbf{V}_A^T\mathbf{W}\mathbf{H}_B\|_F^2
\tag{19}
$$

$$\mathbf{W} \leftarrow \mathbf{W} \odot \frac{\mathbf{V}_A \mathbf{V}_B \mathbf{H}_B^T + \mathbf{V}_A \mathbf{H}_A^T}{\mathbf{V}_A \mathbf{V}_A^T \mathbf{W} \mathbf{H}_B \mathbf{H}_B^T + \mathbf{W} \mathbf{H}_A \mathbf{H}_A^T} \tag{20}$$

$$\mathbf{H}_A \leftarrow \mathbf{H}_A \odot \frac{\mathbf{W}^T \mathbf{V}_A}{\mathbf{W}^T \mathbf{W} \mathbf{H}_A} \tag{21}$$

$$\mathbf{H}_B \leftarrow \mathbf{H}_B \odot \frac{\mathbf{W}^T \mathbf{V}_A \mathbf{V}_B}{\mathbf{W}^T \mathbf{V}_A \mathbf{V}_A^T \mathbf{W} \mathbf{H}_B} \tag{22}$$

The following is the KL divergence chained linkage MUR. The ones matrix in the shape of $V_B$ is found in Equation 23. The ones matrix in Equations 24 and 25 is the shape necessary to allow element-wise division and products.

$$\mathbf{W} \leftarrow \mathbf{W} \odot \frac{V_A \frac{V_B}{V_A^T W H_B} H_B^T + \frac{V_A}{W H_A} H_A^T}{1 H_A^T + V_A 1 H_B^T} \tag{23}$$

$$H_A \leftarrow H_A \odot \frac{W^T \frac{V_A}{W H}}{W^T 1} \tag{24}$$

$$H_B \leftarrow H_B \odot \frac{W^T V_A \frac{V_B}{V_A^T W H_B}}{W^T V_A 1} \tag{25}$$

The general case for Frobenius Loss is (up to the third linkage or $V_D$):

$$\begin{aligned} Loss =& \|\mathbf{V}_A - \mathbf{W}\mathbf{H}_A\|_F^2 + \\ & \|\mathbf{V}_B - \mathbf{V}_A^T \mathbf{W} \mathbf{H}_B\|_F^2 + \\ & \|\mathbf{V}_C - \mathbf{V}_B^T \mathbf{V}_A^T \mathbf{W} \mathbf{H}_C\|_F^2 + \\ & \|\mathbf{V}_D - \mathbf{V}_C^T \mathbf{V}_B^T \mathbf{V}_A^T \mathbf{W} \mathbf{H}_C\|_F^2 \end{aligned} \tag{26}$$

This following is the pattern for any number of linkages. The pattern is expected to continue beyond 3 linkages i.e. past $V_D$. [1]

$$\mathbf{W} \leftarrow \mathbf{W} \odot \frac{... + \mathbf{V}_A \mathbf{V}_B \mathbf{V}_C \mathbf{V}_D \mathbf{H}_D^T + \mathbf{V}_A \mathbf{V}_B \mathbf{V}_C \mathbf{H}_C^T + \mathbf{V}_A \mathbf{V}_B \mathbf{H}_B^T + \mathbf{V}_A \mathbf{H}_A^T}{... + \mathbf{V}_A \mathbf{V}_B \mathbf{V}_C \mathbf{V}_C^T \mathbf{V}_B^T \mathbf{V}_A^T \mathbf{H}_D \mathbf{H}_D^T + \mathbf{V}_A \mathbf{V}_B \mathbf{V}_B^T \mathbf{V}_A^T \mathbf{H}_C \mathbf{H}_C^T + \mathbf{V}_A \mathbf{V}_A^T \mathbf{W} \mathbf{H}_B \mathbf{H}_B^T + \mathbf{W} \mathbf{H}_A \mathbf{H}_A^T} \tag{27}$$

$$\mathbf{H}_A \leftarrow \mathbf{H}_A \odot \frac{\mathbf{W}^T \mathbf{V}_A}{\mathbf{W}^T \mathbf{W} \mathbf{H}_A} \tag{28}$$

$$\mathbf{H}_B \leftarrow \mathbf{H}_B \odot \frac{\mathbf{W}^T \mathbf{V}_A \mathbf{V}_B}{\mathbf{W}^T \mathbf{V}_A \mathbf{V}_A^T \mathbf{W} \mathbf{H}_B} \tag{29}$$

$$\mathbf{H}_C \leftarrow \mathbf{H}_C \odot \frac{\mathbf{W}^T \mathbf{V}_A \mathbf{V}_B \mathbf{V}_C}{\mathbf{W}^T \mathbf{V}_A \mathbf{V}_B \mathbf{V}_B^T \mathbf{V}_A^T \mathbf{W} \mathbf{H}_C} \tag{30}$$

$$\mathbf{H}_D \leftarrow \mathbf{H}_D \odot \frac{\mathbf{W}^T \mathbf{V}_A \mathbf{V}_B \mathbf{V}_C \mathbf{V}_D}{\mathbf{W}^T \mathbf{V}_A \mathbf{V}_B \mathbf{V}_C \mathbf{V}_C^T \mathbf{V}_B^T \mathbf{V}_A^T \mathbf{W} \mathbf{H}_D} \tag{31}$$

If there is a radial branching linkage from matrix $V_B$ that is called $V_E$ then the loss term added would be:

$$\|\mathbf{V}_E - \mathbf{V}_B^T \mathbf{V}_A^T \mathbf{W} \mathbf{H}_E\|_F^2 \tag{32}$$

The chain can then proceed from both $V_E$ and $V_C$. Every time a branch is encountered a similar term is added.

Chained view NMF can be combined with Multiview NMF by creating MUR's according to this equation:

---

[1]Experimentally verified to converge for the first linkage

$$\theta \leftarrow \theta * \frac{\nabla_\theta^- f(\theta)}{\nabla_\theta^+ f(\theta)} \tag{33}$$

$\nabla_\theta^+ f(\theta)$ represents the positive terms in the gradient. $\nabla_\theta^- f(\theta)$ represents the negative terms in the gradient.

## 3   RELATED WORKS

In Ailem et al. (2017) we find an approach to encompass a linked datasets, however this is distinct from our approach and does not incorporate multiple chaining. This is formulated identically in Liu (2019) and called multirelational clustering in Equation 34, again this does not incorporate 2nd or beyond chaining of matrices.

$$Loss = 1/2||\mathbf{V} - \mathbf{WH}||_F^2 + 1/2||\mathbf{M} - \mathbf{WSW}^T||_F^2 \tag{34}$$

Where, V,W,H are the same as above and M is the U matrix in SVD for word vectors, S can be an identity matrix.

NMF using MUR has been an effective method to decompose data matrices for unsupervised feature representations beginning with Seung & Lee (2001). This idea was expanded to multiview NMF that is when there are more than one set of features for a primary key data type Liu et al. (2013). It has been extended to social network community detection as noted in Shi et al. (2015) and Twitter political communities in Ozer et al. (2016). Extending NMF to hypergraphs was done in Wang et al. (2016). Like our proposed approach to find hashtag communities a similar work was performed in He et al. (2017). Shen et al. (2017) analyzes the problem associated with social communities of spammers that spread malicious content using multiview NMF. Yang & Wang (2018) published in IEEE Big Data Mining and Analytics a comprehensive survey on multiview clustering including a section on MultiNMF.

The modern trend has been to incorporate deep learning capabilities especially the attention mechanism as seen in, Chen et al. (2019) adaptation of MultiNMF to the attention paradigm. Additionally the capability to track social media using MultiNMF in the recent COVID-19 epidemic was done by Cruickshank & Carley (2020). Star based schemas Sun et al. (2009) are similar to MultiNMF in that they analyze the data by the primary key, however the star schema is not like a radial tree or cyclical topology that chained NMF is capable of analyzing. Heterogeneous information networks by Sun & Han (2013); Yang et al. (2020) do analyze the higher order topologies however they are concerned with embeddings of the nodes and edges rather than directly analyzing the datapoints of each table and directly optimizing the loss to recreate the each linked table from its matrix factors simultaneously. This simultaneous recreation of all of the datasets from each table's factors and the overall cluster factors is our novel contribution.

## 4   EXPERIMENTS AND CASE STUDIES

### 4.1   SYNTHETIC DATA - LINKAGE INFORMATION PROPAGATION PERFORMANCE, CHARACTERISTICS AND VISUALIZATION

The choice of matrix size in the following experiment section is arbitrary, but the rule when linking matrices is the columns of one must correspond to the rows of the next matrix. In practice this is not a problem because if there is a matrix of words on the rows $\times$ documents in the columns, the matrices that can link are either properties of the documents or the words; for example word vector or document vector matrices.

We created three matrices $V_A$, $V_B$ and $V_C$. The first matrix, $V_A$, is 630 $X$ 350. The rows represent hashtags and the columns geographic locations. The second matrix, $V_B$ has 350 datapoints and 1000 features. The columns or features represent words. So it is basically a graph where each geographic location has predominant words used. Finally the thirds matrix is 1000 $\times$ 20 representing 1000 "words" and 20 features such as word vectors. Notice the matrix multiplication is possible by $630 \times 350 \cdot 350 \times 1000 \cdot 1000 \times 20$, this multiplication is key to our insight.

Table 1: Synthetic Dataset Performance Characteristics

| CHARACTERISTIC | MEAN NMI | STDEV NMI | LINKAGES |
|---|---|---|---|
| Third/Goal Matrix Alone | 0.90 | 0.04 | 0 |
| Second Matrix Alone* | 0.26 | 0.06 | 0 |
| First Matrix Alone** | 0.11 | 0.05 | 0 |
| Second Matrix Chained* | 0.49 | 0.03 | 1 |
| First Matrix Chained** | 0.20 | 0.04 | 2 |

The information that will flow through is from the 7 groups who align with a geographic location, word frequency and word vector style.

We synthesized the first matrix by a sparse matrix with 5% noise, the default sparse value is 0 and the noise is a 1. Then we added random 1's in a specific rectangular pattern along the diagonal, this is visible in the figure and allows the information to flow. If we shuffled the matrix by row these yellow boxes would disappear but if we shuffled the true values identically the matrix would still be useful. It has not been shuffled so that the process can be inspected. The groups can directly be found by performing vanilla NMF or by chaining to a second or even third matrix. The performance increases and the nature of the clusters changes as the complementary and consensus information is factored with chain view NMF. We synthesized a second matrix like the first. Finally, we synthesized the third matrix to have 7 blobs in 20 dimensions, this represents 1000 people in 7 groups measured in 20 different ways.

Table 3 contains the NMI or normalized mutual information score for the different number of linkages 0-2. This was provided by the Scikit-Learn module in Python. [2]

## 4.2 INTERPRETATION OF LINKED MATRIX INFORMATION PROPAGATION AND VISUALIZATION

To understand how information flows through the data tables we illustrated the dot products as they appear in the equations above and plot them in 3 dimensions on a synthetic Gaussian grouping as follows in Figure 3 and Table 2. In Figure 3 we are viewing whether clusters remain intact after chaining. While a real dataset would not be as well behaved, we can still demonstrate the preservation of clear clusters through matrix multiplication in the loss functions. For instance, $\mathbf{V}_C^T\mathbf{V}_B^T\mathbf{V}_A^T\mathbf{W}\mathbf{H}_C$ approximates $\mathbf{V}_D$, where $\mathbf{V}_A...\mathbf{V}_D$ are simply relational database tabular matrices. When $\mathbf{V}_A...\mathbf{V}_C$ are multiplied together and then multiplied with the overall factor matrix $\mathbf{W}$ and the factor matrix $\mathbf{H}_C$ we recreate $\mathbf{V}_D$. Thus, the decomposition of the relational database can be understood to preserve clusters that will occur as large row or column values in the factor matrices. We will prove that the approximation products don't just allow for the same dimensions.

$$V_A = WH_A \tag{35}$$

$$V_B = V_A^T W H_B \tag{36}$$

by substitution:

$$V_B = H_A^T W^T W H_B \tag{37}$$

From this we observe $W^T W$ is the Gram matrix, in data science this is a linear kernel that measures similarity and has rows and columns equal in cardinality to the number of clusters in NMF. Since we are multiplying the column factors of $V_A$ stored in $H_A$ by the similarity of the clusters by the column factors of $V_B$ stored in $H_B$. This gives us the columns of $V_A$ and the columns of $V_B$ aka the rows and columns of $V_B$, thus proving that our approximations are adequate or that Equation 37 is reasonable.

---

[2]* and ** are significantly different at a level of 0.001

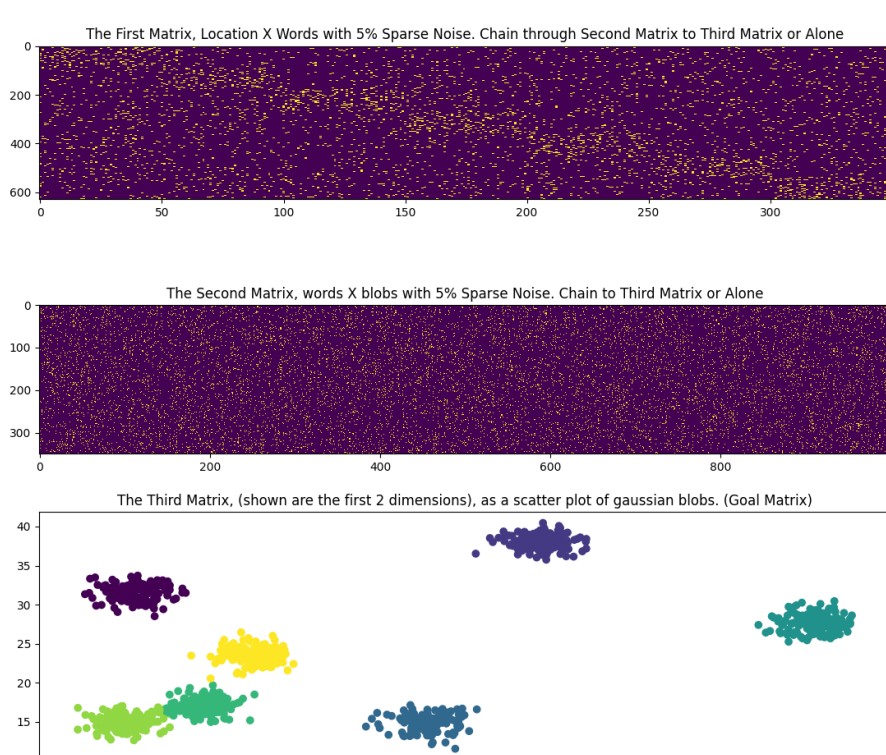

Figure 2: Chaining together three matrices, $1000\times20$ to $350\times1000$ to $630\times350$. This chain could also have been done on real data for instance: hashtag $\times$ location graph to location $\times$ word topic graph finally to a word topic $\times$ continuous feature blob dataset. Chained views improve the ability to discern clusters in each of the linking matrices and give a unique interpretation that depends on which information or matrices are included in the chain. (In the blobs, we show the first 2 dimensions of the 20 dimensional dataset.)

| Mean Input 1 (Matrix 1) | Mean Input 2 (Matrix 2) | Arithmetic Output Mean (Inner Product) |
|---|---|---|
| 0.8 | 3 | 20 * 0.8 * 3.0 = 48 |
| 0.8 | 5 | 20 * 0.8 * 5.0 = 80 |
| 2.8 | 3 | 20 * 2.8 * 3.0 = 168 |
| 2.8 | 5 | 20 * 2.8 * 5.0 = 280 |

Table 2: Truth table for multidimensional input signals to output modeling the flow of information for NMF clustering of chained data. The number of features in matrix 1 is 20.

## 4.3  KL DIVERGENCE CHAINED-JAX VS VANILLA KL FOR 50 AUTHORS

Chained NMF has more usefulness than multiview NMF that cannot handle the case of document$\times$word and word$\times$word vector. We incorporated word vectors with chained view using JAX on the 50 authors C50 dataset. We used the Google 300 dimension word vectors all thresholded at 0 so that the data was non-negative.

Additionally, we analyzed the 20 Newsgroups using chained link KL with word vectors vs Vanilla KL and observed a slight decrease in NMI and ARI but better clusters overall found in Table 4.

Table 3: C50 Dataset Performance Characteristics

| Cluster Mode and Metric | RUN 1 | RUN 2 | RUN 3 | RUN 4 | P-VALUE |
|---|---|---|---|---|---|
| KL Chained-Jax and Word Vectors NMI | 0.46 | 0.45 | 0.47 | 0.49 | $< 0.01$ |
| KL Vanilla NMI | 0.42 | 0.42 | 0.40 | 0.41 | See prior row |
| KL Chained-Jax and Word Vectors ARI | 0.18 | 0.17 | 0.18 | 0.20 | $< 0.05$ |
| KL Vanilla ARI | 0.17 | 0.16 | 0.14 | 0.14 | See prior row |

## 5 ALGORITHM TIME AND SPACE COMPLEXITY ANALYSIS

Inspection of the gradient shows four salient aspects. The first is the radial structure of the input, this is an factorial type of complexity. Secondly, because we are factoring a radial explosion of matrices, if in data science there is a law of data fusion similar to the law of large numbers in statistics then we can say that if there are enough matrices factored then there are certain aspects of hidden knowledge that change less and less the more complementary and consensus tables are factored. This invariance in hidden knowledge discovery is strong motivation to use this method despite its poor complexity analysis for the analytic solutions. Fourth is that the autodifferentiation is not necessarily performing all of the operations that would be done in an analytical solution. While our method performing even modest sized matrix operations will quickly run into TB of memory and computational decades on high performance machines we can quickly calculate the analytic terms of the solution using autodifferentiation. Empirically, we observed errors diminishing monotonically using the target matrix and a linking matrix, we see no reason for third matrix and beyond links to perform differently as they are based upon the same principle. A future work would involve computational proofs because we use the automatic gradient solvers for efficiency. Fourth is that all matrix operations are parallelizable.

## 6 DISCUSSION

The hope of this work is to allow a researcher to collocate all of the relevant data into the model and see what is previously unknown and potentially useful. Their future task will be what a consensus of knowledge can produce, with the hope of a deeper understanding of phenomena that cannot be attained using any single or pair of source data. Artificial intelligence approaches to incorporating a wide range of data has the potential to transform what is considered the lack of algorithmic common sense reasoning to something approaching the ability of human reasoning. We are motivated to extend our methods by research that discovers emergent social networks such as Yang et al. (2013), this is complementary method to ours that analyzes graphical topologies and features of nodes, however we extended to the varying features and items of a relational database but would like to explore incorporation of the graphical parameters into our method. Additionally, the work by Gong et al. (2013) shows a unique particle swarm methodology that integrates complex networks for clustering, we would like to incorporate multiobjective optimization into our decomposition of relational tables, particle swarm has proven many times to be competitive in results and would be a promising future direction of research.

Multiplying chained matrices together mimics the principle of dilution, for instance if a large matrix multiplies a very small one then most of the information is lost going into the smaller, or at least it becomes more information dense. Appropriately, there is nowhere to store the data in the smaller matrix product. Also large matrix to large matrix allows the flow of information readily. And in the third case a small matrix does not have much information to pass through to a larger matrix.

## 7 REPRODUCIBILITY STATEMENT

In the appendix we provide the code used to compare the vanilla NMF to chained link NMF, we however leave the TF-IDF implementation out because it was from Scikit-Learn using Python 3.

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

# A APPENDIX

## A.1 CAVEATS

Weakness of this method: The parameters must match exactly to "chain" together tables. However it should always be possible to include all of the attributes of a relational database to find an embedding of a target feature.

## A.2 PROOFS AND MUR RULES IN CODE FOR REPRODUCING RESULTS

```
import jax.numpy as jnp
from jax import grad

###########Algorithm for Chained Link NMF using KL and Matrix Operations
        for epoch in range(200):
            num = np.divide(Y,np.linalg.multi_dot((X.T,W,H2)))
            num = np.linalg.multi_dot((X,num,H2.T))
            num += np.divide(X,W@H)@H.T
            den = np.ones((W.shape[0],H.shape[1]))@H.T
            den += np.linalg.multi_dot((X,np.ones(Y.shape),H2.T))
            den[den == 0] = eps_machine
            W *= num / den

            num = W.T@np.divide(X,W@H)
            den = W.T@np.ones((W.shape[0],H.shape[1]))
            den[den == 0] = eps_machine
            H *= num / den

            num = np.divide(Y,np.linalg.multi_dot((X.T,W,H2)))
            num = np.linalg.multi_dot((W.T,X,num))
            den = np.linalg.multi_dot((W.T,X,np.ones((X.shape[1],H2.shape[1]))))
            den[den == 0] = eps_machine
            H2 *= num / den

###########Algorithm for Chained Link NMF using KL and JAX
    t1 = lambda X, W, H: jnp.sum(jnp.multiply(X, jnp.log(jnp.divide(X,W@H))))
    t2 = lambda X: jnp.sum(-X)
    t3 = lambda W, H: jnp.sum(W@H)
    t4 = lambda Y, X, W, H2: jnp.sum(jnp.multiply(Y, jnp.log(jnp.divide(Y,X.T@W@H2)))
    t5 = lambda Y: jnp.sum(-Y)
    t6 = lambda X, W, H2: jnp.sum(X.T@W@H2)

    terms = [t1,t2,t3,t4,t5,t6]
    Args = lambda: [(X,W,H),(X,),(W,H),(Y,X,W,H2),(Y,),(X,W,H2)]
    Windex = [1,None,0,2,None,1]
    Hindex = [2,None,1,None,None,None]
    H2index = [None,None,None,3,None,2]

     MU(Xindex):
         args = Args()
         grads = [grad(t,argnums=i)(*a) for a,i,t in zip(args,Xindex,terms) if i is no
         numerator = [np.abs(g) for g in grads if np.any(g < 0)]
         numerator = np.sum(numerator,axis=0)
         denominator = [g for g in grads if np.any(g > 0)]
         denominator = np.sum(denominator,axis=0)
         return  numerator / (denominator + eps_machine)
```

```
    for epoch in range(200):
        W *= MU(Windex)
        H *= MU(Hindex)
        H2 *= MU(H2index)

###########Algorithm for Chained Link NMF using Frobenius and
###########JAX or Matrix Operations
        '''
        min ||X-WH|| + ||Y-XTWH2|| s.t. W, H, H2 >= 0

        tr((X-WH)T (X-WH))
        tr((Y-XTWH2)T (Y-XTWH2))
        '''
        t1 = lambda X: jnp.trace(X.T@X)
        t2 = lambda X, W, H: -jnp.trace(X.T@W@H)
        t3 = lambda X, W, H: -jnp.trace(H.T@W.T@X)
        t4 = lambda W, H: jnp.trace(H.T@W.T@W@H)

        t5 = lambda Y: jnp.trace(Y.T@Y)
        t6 = lambda Y, X, W, H2: -jnp.trace(Y.T@X.T@W@H2)
        t7 = lambda Y, X, W, H2: -jnp.trace(H2.T@W.T@X@Y)
        t8 = lambda X, W, H2: jnp.trace(H2.T@W.T@X@X.T@W@H2)

        for epoch in range(200):
            Wnum = X@H.T + np.linalg.multi_dot([X,X.T,W,H2,H2.T])
            Wden = np.linalg.multi_dot([W,H,H.T]) + np.linalg.multi_dot([X,Y,H2.T])
            Wden[Wden==0] = eps_machine
            #W *= MU(Windex)
            W *= Wnum/Wden

            Hnum = W.T@X
            Hden = np.linalg.multi_dot([W.T,W,H])
            Hden[Hden == 0] = eps_machine
            H *= Hnum/Hden
            #H *= MU(Hindex)

            H2num = np.linalg.multi_dot([W.T,X,Y])
            H2den = np.linalg.multi_dot([W.T,X,X.T,W,H2])
            H2den[H2den == 0] = eps_machine
            u1 = H2num / H2den
            #u2 = MU(H2index)

##Basic Procedure for Testing AutoDiff Gradient Matches Matrix Operation
##(Proof of Correctness)

from jax import grad
import jax.numpy as jnp
import numpy as np
X=np.random.normal(0,1,(15,20))
Y=np.random.normal(0,1,(20,25))
W=np.random.normal(0,1,(15,7))
H=np.random.normal(0,1,(7,20))
H2=np.random.normal(0,1,(7,25))
X,Y,W,H,H2 = [np.abs(x) for x in [X,Y,W,H,H2]]
t4 = lambda Y, X, W, H2: jnp.sum(jnp.multiply(Y, jnp.log(jnp.divide(Y,X.T@W@H2))))
g1 = grad(t4,argnums=3)(Y,X,W,H2)
num = np.divide(Y,np.linalg.multi_dot((X.T,W,H2)))
num = np.linalg.multi_dot((W.T,X,num))
g2 = num
```

```
if np.allclose(g1,-g2):
    print("H2 Num is good.")
```

### A.3 RESEARCH FRONTIERS IN MINING HETEROGENEOUS INFORMATION NETWORKS

Quoting from Sun & Han (2013), "Viewing interconnected data as an information network and studying systematically the methods for mining heterogeneous information networks is a promising frontier in data mining research... How to discover such hidden networks and how to mine knowledge (e.g., clusters, behaviors, and anomalies) from such hidden but non-isolated networks (i.e., still intertwined with the gigantic network in both network linkages and semantics) could be an interesting but challenging problem." This latest NMF algorithm extended from MultiNMF is proposed to be called chained NMF. We believe the greatest discoveries are yet to be found and this papers impact in this interesting and challenging problem pushes the envelope of the possible and future research will no doubt unlock doors to finding hidden knowledge in data by analyzing entire volumes of data that have a wide variety of types.

### A.4 MATRIX PRODUCT CLUSTER PRESERVATION VISUALIZATION

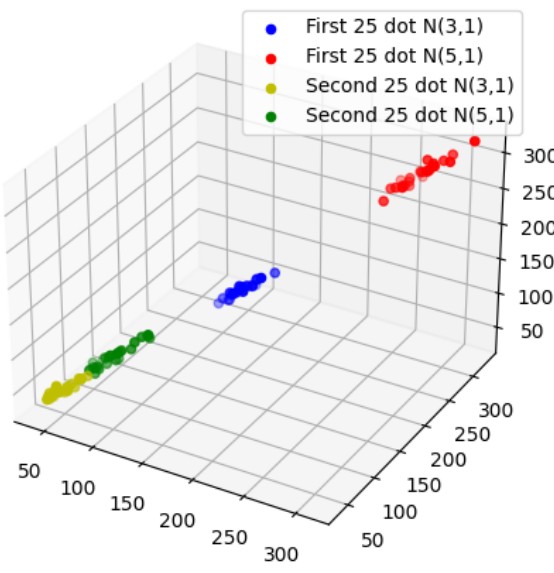

Figure 3: Chaining visualization on a controlled synthetic dataset. Notice the clusters easily visualized corresponding to varying the input from either matrix, see Table 2

### A.5 REASONABLE IMPROVEMENTS IN QUALITY OF CLUSTERS USING CHAINED NMF ON NEWSGROUPS.

Typically the best clusters are achieved using KL Divergence Loss for NMF with MUR. We compare them qualitatively to chained NMF using Google 300 dimension wordvectors for 7 of the newsgroups in the table 4.

Table 4: 7 Newsgroups Characteristic Cluster Comparison (subset of 20 Newsgroups)

| | Chained Link NMF KL 7 Cluster Word Choices for 7 Newsgroups |
|---|---|
| talk.politics.mideast | israel, israeli, government, said, state, palestinian, part, arab, war, during, peace, against, back, year, children, off, agree, soldiers, american, troops |
| sci.electronics | power, circuit, battery, edu, voltage, light, space, current, radio, sky, electronics, output, thanks, 10, energy, information, msg, tv, off, temperature |
| sci.med | she, her, medical, doctor, disease, pain, patients, treatment, cancer, go, food, cause, drug, medicine, ago, patient, really, effects, problem, wife |
| sci.space | space, launch, ground, money, msg, nasa, cost, moon, high, radar, re, low, costs, dc, funding, still, safety, buy, long, real |
| sci.crypt | key, chip, clipper, encryption, government, phone, keys, system, nsa, algorithm, law, security, escrow, des, number, public, crypto, data, bit, secure |
| comp.graphics | thanks, graphics, program, please, image, file, files, software, mail, looking, ftp, space, format, hi, advance, images, help, gif, pc, windows |
| soc.religion.christian | god, jesus, church, jews, christian, believe, christ, christians, bible, israel, him, armenian, faith, jewish, armenians, sin, world, life, turkish, christianity |
| | Vanilla NMF KL 7 Cluster Word Choices for 7 Newsgroups |
| talk.politics.mideast | 1993, during, april, armenians, argic, 10, armenian, between, 93, ground, 000, center, last, 12, 30, 19, against, city, air, russian |
| sci.electronics | better, company, around, few, enough, actually, buy, back, going, re, cost, big, 10, off, build, go, long, probably, got, down |
| sci.med | edu, cause, banks, back, case, cadre, doctor, effect, ago, gordon, medical, chastity, disease, help, soon, anything, deleted, intellect, day, geb |
| sci.space | thanks, please, mail, advance, space, email, book, nasa, looking, another, information, anybody, articles, look, ll, contact, able, around, copy, help |
| sci.crypt | clipper, chip, key, government, encryption, system, algorithm, public, keys, data, com, bit, phone, nsa, security, using, chips, secret, already, anything |
| comp.graphics | thanks, graphics, file, available, looking, please, files, ftp, program, image, advance, help, hi, appreciated, 3d, bit, code, format, etc, color |
| soc.religion.christian | god, believe, christian, israel, him, church, again, against, come, both, fact, christians, christ, between, agree, another, bible, said, jesus, however |

