# OpenReview forum: "Chaining Data - A Novel Paradigm in Artificial Intelligence Exemplified with NMF based Clustering"
_ICLR.cc/2022/Conference — ICLR 2022 Submitted_

### Official Review · Reviewer_k2WG · 2021-10-28

**Correctness:** 3
**Technical Novelty And Significance:** 3
**Empirical Novelty And Significance:** 2
**Recommendation:** 5
**Confidence:** 4

**Details Of Ethics Concerns:**

No ethical concerns with this paper.

**Main Review:**

__Strong points__: Overall, the paper’s strengths are the technical novelty of the proposed chain-view NMF and the impact of the problem it is addressing.

- The paper presents a new technique for dealing with a real-world data problem that is straightforward in its technical grounding. The extensions the paper proposes to NMF are both conceptually straightforward and a rather elegant way of attacking the data scenario of linked and multi-view data
- The problem of finding good representations of phenomena, when those phenomena can be described by both multi-view and linked data, is a significant problem. Being able to successfully do this task can lead to better outcomes in fundamental machine learning tasks like clustering.

__Weak points__: Overall, the paper’s weaknesses are in its clarity and its empirical significance.

- The paper has some serious clarity issues that detract from the reader being able to understand the paper and contributions of the paper. The paper needs to be reviewed for grammar and typos, as they are frequent and do detract from understanding the text. One example is at the bottom of page 1, the second paragraph of the introduction, wherein the sentence “A multiNMF is where…” is repeated. Another example is in section 2.1, where it looks like from the text that “Hashtags _X_ tf-idf …” implies that that matrix should be $X$, but it is instead $V$ in the subsequent equation.
Also, within the proposed contribution of section 2.3, while it is clear how to deal with the situation of linked data (by multiplying those $V_*$ matrices into the second part of the loss function, it's not clear how to deal with multi-view data (where you have two or more data sets, or tables, with the same row entities). Do you just add an extra term (same as the original term, but with the W being the same across terms) to the loss function for this other table? If so, how do you control for the different impacts of the added together parts of the loss function (i.e. would you weight the different sub-components of the loss function)? This also leads to the intriguing idea of how one should model their data relationships to get the best cluster structure, especially when given the design choice of whether to make a certain table of data chained or multi-view. Also, the intuition idea of chaining the matrices by the cartesian product should be explained. While it certainly makes the matrix math work out, why is that the best way to propagate information from those matrices?
Figure 2, while a neat image, really doesn’t seem to contribute much actual information to the paper, since the visual inspection isn’t stated as an empirical evaluation test. This space would be better used for either more explanation on the empirical setup (more on that later) or doing more empirical tests.

- The empirical investigations are unclear as to how the experiments were set up and evaluated For example, what clustering was done for the AMI and ARI values? And, why is the synthetic test set up different than the real-world data set up? It seems like both should share the same test setup. For example, why does Table 2 look at NMI and ARI, while Table 1 does not and why does Table 1 consider various different ways of chaining and Table 2 does not? Finally, the word vectors in 4.3, is truncated the negative values really the best way to make sure those vectors are positive? Won’t this cause a loss of critical information, since they are embeddings in a real-valued space (implying negative values contain information)? Perhaps a better way would be to do a transformation of them into a positive space by scaling and/or normalization.
The empirical investigations, from Table 2, really do not show much performance improvement over the KL-optimized standard NMF, especially when one considers the chained version requires so many more matrix operations than just a single view NMF.

- Finally, the paper overstates its claim about the proposed technique being a novel paradigm in artificial intelligence. The idea of using more than one view of data to understand phenomena is the foundation of multi-view machine learning, and the idea of using linked data underlies techniques in Heterogeneous Information Networks and complex network clustering. So, while it is a new technique, the paper does not present a new paradigm.


**Summary Of The Paper:**

The paper presents a new technique, grounded in non-negative matrix factorization (NMF), for unsupervised, representation learning of linked data. The paper presents a technique that can deal with data scenarios whereby a certain phenomenon of interest has multiple views to characterize that behavior (i.e. traditional multi-view clustering set up), but also the data in those views can be linked to further data, as in a relational data.

**Summary Of The Review:**

I recommend the paper be accepted. While it does need some serious work in clarity and maybe some additional empirical investigation, it does present a new technique that is attacking a meaningful problem.

---

> ### Author Response · Authors · 2021-11-21
> **Address to concerns,questions, improvements, weaknesses and clarity.**
>
> Clarity:We completely reviewed the paper for inconsistencies in writing, grammar and clarity. The paper was edited to replace the unclear with plain language that is more readily understood.  Additionally we have revised our notation. We apologize for the inconsistent writing and use of undefined notation.
>
> Empirical significance:Due to time, space and data constraints we were not able to perform analysis, however we added a subfigure to Figure 1 that explains a heterogeneous database task. This method differs from graph embeddings because entire relational tables factors can reconstruct the nodes in the graph.  (Bottom) Depicted is a fictitious relational dataset model relating to black holes where arrows represent primary keys that are not provided.  We propose performing analysis at the NMF insertion point Sagittarius A* analyzing at the datapoint level, this novel approach is different than analyzing the graph structure with graph embeddings or only the immediate neighboring tables. Clusters, factors and other NMF outputs are not shown, but only the input structure in all three of the subfigures.
>
> would you weight the different sub-components of the loss function)?
> Thank you for this interesting observation. It may be important to decrease the loss assigned to increasingly complex chains so they do not contribute to W the same as the most proximal tables. Alternatively one may solve for the last term in the loss function and by doing so incorporate all of the data matrices. Solving for the last term does have one drawback, it disproportionately weighs all data matrices equal, this may not be the intent. It may be necessary to experiment with different weightings or scaling the input matrices.
>
> Design choice of whether to make a certain table of data chained or multi-view. Unfortunately, the two different choices are governed by the topology of the data rather than the designer’s choice.
>
> While it certainly makes the matrix math work out, why is that the best way to propagate information from those matrices? It is not clear that we used the cartesian product, rather we believe that we used the dot product to propagate information, in Figure 3 we show how the dot product carries information from the nonnegative data, in a sense it resembles a capture of the signal in the data and when signal is present then the dot product moves the information further outward from zero.  Please see equations 35, 36 and 37 that have been added to show why the math works out.
>
> This space would be better used for either more explanation on the empirical setup (more on that later) or doing more empirical tests: We removed this image to make more room for reviewer comments, observations, questions and discussion.
>
> For example, why does Table 2 look at NMI and ARI, while Table 1 does not and why does Table 1 consider various different ways of chaining and Table 2 does not? While we would like to produce a consistent comprehensive testing suite or use a previously created one, we only produced a proof of concept, in a second submission to a journal we will produce more real world testing to show the broader applicability and unique advantages of the method compared to other approaches such as HNEs.
>
> Perhaps a better way would be to do a transformation of them into a positive space by scaling and/or normalization. We believe adding additional vectors and then truncating them provides a more appropriate data source because a negative value may take on a different meaning than a positive value. If scaled a word vector would have both the positive and the negative meaning combined, then we may have unexpected results. However experimentally the scaling may prove to be more correct, we will determine this given additional time for a second submission to a journal.
>
> Single view NMF: The performance is primarily to show how well it clusters to known groups, this may not be indicative of cluster quality. There are multiple good clusterings of a dataset and comparing to a known clustering gives only a partial picture, but is better than a qualitative only assessment. The algorithmic complexity has been added to section 5.
>
> Heterogeneous Information Networks and complex network clustering. So, while it is a new technique, the paper does not present a new paradigm:  yang2020heterogeneous  present 13 popular heterogeneous network embedding (HNE) algorithms of which ours is a type as we produce node embeddings but not link embeddings. Furthermore, they give a standard by which embeddings of nodes in a graph or links by type can compare HNE’s in a semisupervised setting. However this is where our work diverges; we take a completely unsupervised approach and embeddings are directly interpretable. A unique advantage of our approach is we evaluate the entire table in a primary keyed table, rather than the topology of a graph and its embedding.

---

> > ### Comment · Reviewer_k2WG · 2021-11-23
> > **Comments after reading the updated version**
> >
> > Dear Authors,
> >
> > I appreciate the work updating the manuscript. The paper's claims and its place within the existing literature is much improved and it better highlights the contribution of the paper. I also appreciate adding in the interpretation section on the propagation of the signal through chaining. I do think there are still some important clarity and correctness issues that have not been addressed. For one, the second paragraph of the introduction seems out of place. It seems like this paragraph would be better placed in the methods section. Second, I don't understand the reasoning provided in the reply about why the paper truncates the word embeddings in the experimental section. The reply makes it sound as though removing negative values from the embeddings (truncating them) removes negative word sentiment from the word embedding. This doesn't seem correct as the coordinates in the latent space do not necessarily correlate to any sentiment of a word (unless all negative sentiment words happen to get mapped to a negative sector of the latent space by the embedding method), and because you would still want to capture that sentiment as part of the meaning of the word. I understand that you can't have negative values in a Non-negative matrix factorization, but I do not think just simply substituting negative values with zero is the correct approach. Finally, the tests against the synthetic data set in section 6.1 are still unclear. It's not immediately clear how the three matrices are chained together or what clustering technique was used on the NMF-derived embeddings. And, from my read of table 1, it looks like chaining matrices produce a much worse NMI than just clustering on the goal matrix alone, which would recommend against doing any kind of chaining. If this interpretation is correct, there needs to be some explanation of this result. I don't believe the paper is ready for publication until these clarity issues are addressed.

---

### Official Review · Reviewer_QKqU · 2021-11-01

**Correctness:** 2
**Technical Novelty And Significance:** 2
**Empirical Novelty And Significance:** 1
**Recommendation:** 3
**Confidence:** 4

**Main Review:**

Strength
- The authors studied the ease of real-world application. They discussed the viability of using commonly available auto-differentiation tools (JAX).
- The use of two different kinds of dataset to test the algorithm: handwriting and documents.

Weakness
- Writing needs substantial improvements. It's generally hard to follow the paper. For example there are sudden turns of topic (Abstract: "..it is commonplace to identify hashtag ..."), and very easy to spot errors (Introduction, 2nd paragraph, "A multiNMF is where the primary key ..." is repeated).
- Section 2.1. The use of hashtag and words is sudden, Reviewer expect some introduction on the task: recommendation or prediction?
- Equation 1, what is V?
- Section 4.3. "We attained the following results" but the authors continued to discuss experiment set up, not the results.
- There are many other similar cases of writing inconsistency. Reviewer won't list them all.


**Summary Of The Paper:**

The paper studied clustering of data that has multiple dimensions of relation. It proposed a chained non-negative matrix factorization (NMF) technique, which, according to the authors, is different from existing multiple-view NMF. Chained NMF can model data that has chained relations (much like external key in relational database) and produce better clusters.

The authors evaluated Chained NMF with synthetic data that seems to resemble documents with hashtags and authors. NMI and ARI are both improved. Another illustrative evaluation was done on handwriting characters data.


**Summary Of The Review:**

The paper discussed a way to cluster a dataset with multiple chainable joining keys. The authors provided mathematical discussion and evaluation on a small synthetic dataset. However, the paper itself is poorly written, to a point that it's hard for Reviewer to understand how the authors arrived at the main idea and what its strength and weakness are. Clarity is the main weakness.

---

> ### Author Response · Authors · 2021-11-21
> **A brief reply to all of the concerns and weaknesses given.**
>
> Another illustrative evaluation was done on handwriting characters data.: Not true, although we wish it were, this is building up to the main contribution and showing what is already existing in the literature. Likewise in the comment, “The use of two different kinds of dataset to test the algorithm: handwriting and documents.” we did not analyze handwriting but we analyzed C50 authorship for NMI and ARI, 20 newsgroups for qualitative differences and 2 different synthetic datasets to illustrate how information flows through linked datasets. We also removed the handwriting dataset to allow for more analysis and discussion of the proposed method on request of a reviewer and our agreement that it would improve the paper.
>
> Writing Inconsistency: A significant rewrite of the abstract, introduction was done for writing consistency and clarity.
>
> Clarity is the main weakness: We completely reviewed the paper for inconsistencies in writing, grammar and clarity. The paper was edited to replace the unclear with plain language that is more readily understood.

---

> > ### Comment · Reviewer_QKqU · 2021-11-23
> > **Comments after reading the updated version.**
> >
> > Thanks for writing an updated version! I can see that abstract and certain sections are improved.
> >
> > However, the clarity issues still exist. For example, I had a hard time trying to understand your purpose of the second paragraph of Introduction. It's also not clear what the discussion part want to convey. Another example is that you named word matrix as H and hashtag matrix as W? Maybe you want to say that W = weight? But it could be confusing for readers to follow, at least for me. This is not a exhaustive list of course but you can see my point about clarity.
> >
> > I feel that the idea in the paper is still interesting, but the authors need to articulate it better to make sure it's clear and valuable to the readers.

---

### Official Review · Reviewer_RkXc · 2021-11-01

**Correctness:** 3
**Technical Novelty And Significance:** 2
**Empirical Novelty And Significance:** 2
**Recommendation:** 5
**Confidence:** 4

**Main Review:**

While the idea of incorporating star-schema information is not new, this paper proposes a new approach to model the non-multi-view information. There have been several works in heterogeneous networks such as NetClus [1] and CESNA [2] where the heterogeneous information sources (feature tables) have been clustered. It is not clear how the chained NMF modeling improves over the existing clustering models in the prior literature.

I believe more work needs to be done to understand the convergence of the algorithm, the time and space complexity analysis. Further, more real-world applications should be tried like heterogeneous database-based tasks to understand the utility of chained NMF modeling.


[1] Sun, Yizhou, Yintao Yu, and Jiawei Han. "Ranking-based clustering of heterogeneous information networks with star network schema." Proceedings of the 15th ACM SIGKDD international conference on Knowledge discovery and data mining. 2009.
[2] Yang, Jaewon, Julian McAuley, and Jure Leskovec. "Community detection in networks with node attributes." 2013 IEEE 13th international conference on data mining. IEEE, 2013.

**Summary Of The Paper:**

This paper proposes a linked NMF based clustering to tackle the problem of linked dataset. The authors shows that chained NMF provides new way to incorporate information and interpret results in comparison to the multi view clustering approaches which are limited to the same entities across tables (or different views of the same entity). Experimentally, the authors show that chained NMF perform better than vanilla NMF at clustering Newsgroups.

**Summary Of The Review:**

While this paper provides novel idea, it lacks some comparison and justification in comparison to the existing literature. More work needs to be done to justify the modeling choice. Also, more experimental work needs to be done to justify the utility of the above model. In light of the above observations, I have made my recommendation of the paper.

---

> ### Author Response · Authors · 2021-11-21
> **Reply to Comments, Observations, Clarifications, Requests and Questions.**
>
> Clarity: We completely reviewed the paper for inconsistencies in writing, grammar and clarity. The paper was edited to replace the unclear with plain language that is more readily understood.
>
> Star Schema: Thank you for this suggestion we mentioned star based schema throughout the paper and compared it to our method and we believe our contribution is reasonable because it includes these additional links and factors entire relational databases Additionally it could be stated that we are concerned with features of features in other words higher order features beyond the first link.
>
> CESNA: We are motivated to extend our methods by research that discovers emergent social networks such as \cite{yang2013community}, this is complementary method to ours that analyzes graphical topologies and features of nodes, however we extended to the varying features and items of a relational database but would like to explore incorporation of the graphical parameters into our method.
>
> Convergence of the algorithm, the time and space complexity analysis: Thank you for this comment, we took advantage of this suggestion to strengthen our argument as posed the following explanation: Inspection of the gradient shows four salient aspects. The first is the radial structure of the input, this is an factorial type of complexity. Secondly, because we are factoring a radial explosion of matrices, if in data science there is a law of data fusion similar to the law of large numbers in statistics then we can say that if there are enough matrices factored then there are certain aspects of hidden knowledge that change less and less the more complementary and consensus tables are factored. This invariance in hidden knowledge discovery is strong motivation to use this method despite its poor complexity analysis for the analytic solutions. Fourth is that the autodifferentiation is not necessarily performing all of the operations that would be done in an analytical solution. Empirically, we observed errors diminishing monotonically using the target matrix and a linking matrix, we see no reason for third and beyond links to perform differently as they are based upon the same principle. A future work would involve computational proofs because we use the automatic gradient solvers for efficiency. Fourth is that all matrix operations are parallelizable.
>
> Real-world applications should be tried like heterogeneous database-based tasks: Due to time, space and data constraints we were not able to perform analysis, however we added a subfigure to Figure 1 that explains a heterogeneous database task. This method differs from graph embeddings because entire relational tables factors can reconstruct the nodes in the graph.  (Bottom) Depicted is a fictitious relational dataset model relating to black holes where arrows represent primary keys that are not provided.  We propose performing analysis at the NMF insertion point Sagittarius A* analyzing at the datapoint level, this novel approach is different than analyzing the graph structure with graph embeddings or only the immediate neighboring tables. Clusters, factors and other NMF outputs are not shown, but only the input structure in all three of the subfigures.
>
>
> Lacks some comparison and justification in comparison to the existing literature: We added more information in section 3 starting at: . Star based schemas Sun et al. (2009) are similar to MultiNMF in that they analyze the data by the primary key, however the star schema is not like a radial tree or cyclical topology that chained NMF is capable of analyzing.  Heterogeneous information networks by Sun & Han (2013); Yang et al. (2020) do analyze the higher order topologies however they are concerned with embeddings of the nodes and edges rather than directly analyzing the datapoints of each table and directly optimizing the loss to recreate the each linked table from its matrix factors simultaneously. This simultaneous recreation of all of the datasets from each table’s factors and the overall cluster factors is our novel contribution.
>
> More work needs to be done to justify the modeling choice: Please see section 4.2 and Equations 35, 36 and 37.
>
> Also, more experimental work needs to be done to justify the utility of the above model: We analyzed C50 authorship for NMI and ARI, 20 newsgroups for qualitative differences and 2 different synthetic datasets to illustrate how information flows through linked datasets.  Due to space and time constraints we did not analyze additional data.
>
> The model choice was to accommodate the fusion of data in relational databases, while graphs and their table-like form of adjacency matrices have been done before, to our knowledge the application of a familiar table format linked by keys has not specifically been optimized for clustering.

---

### Decision · Program_Chairs · 2022-01-20

**Decision:**

Reject

**Comment:**

The paper proposes a new approach for linked-view clustering based on chained non-negative matrix factorization. Reviewers highlighted that paper proposes a novel and interesting approach to an important problem. However, reviewers raised also significant concerns regarding clarity of presentation (motivation, general approach, contributions, scope) as well as the experimental evaluation. Reviewers raised also concerns regarding justification of the approach being a novel paradigm. After author response and discussion, all reviewers and the AC agree that the paper is not yet ready for publication due to the aforementioned issues.